# A Wide Spectrum of Genetic Disorders Causing Severe Childhood Epilepsy in Taiwan: A Case Series of Ultrarare Genetic Cause and Novel Mutation Analysis in a Pilot Study

**DOI:** 10.3390/jpm10040281

**Published:** 2020-12-15

**Authors:** Syuan-Yu Hong, Jiann-Jou Yang, Shuan-Yow Li, Inn-Chi Lee

**Affiliations:** 1Division of Pediatric Neurology, Department of Pediatrics, Children’s Hospital, China Medical University Hospital, Taichung 40447, Taiwan; D28320@mail.cmuh.org.tw; 2Genetics Laboratory and Department of Biomedical Sciences, Chung Shan Medical University, Taichung 40201, Taiwan; jiannjou@csmu.edu.tw (J.-J.Y.); syl@csmu.edu.tw (S.-Y.L.); 3Division of Pediatric Neurology, Department of Pediatrics, Chung Shan Medical University Hospital, Taichung 40201, Taiwan; 4Institute of Medicine, School of Medicine, Chung Shan Medical University, Taichung 40201, Taiwan

**Keywords:** WES, infantile epilepsy, genetic variation, personal therapy

## Abstract

Background: Pediatric epileptic encephalopathy and severe neurological disorders comprise a group of heterogenous diseases. We used whole-exome sequencing (WES) to identify genetic defects in pediatric patients. Methods: Patients with refractory seizures using ≥2 antiepileptic drugs (AEDs) receiving one AED and having neurodevelopmental regression or having severe neurological or neuromuscular disorders with unidentified causes were enrolled, of which 54 patients fulfilled the inclusion criteria, were enrolled, and underwent WES. Results: Genetic diagnoses were confirmed in 24 patients. In the seizure group, *KCNQ2, SCN1A, TBCID 24, GRIN1, IRF2BPL, MECP2, OSGEP, P*ACS1, *PIGA, PPP1CB, SMARCA4, SUOX*, SZT2, *UBE3A*, 16p13.11 microdeletion, [4p16.3p16.1(68,345–7,739,782)X1, 17q25.1q25.3(73,608,322–81,041,938)X3], and *LAMA2* were identified. In the nonseizure group, *SCN2A, SPTBN2, DMD*, and *FBN1* were identified. Ten novel mutations were identified. The recurrent genes included *SCN1A, KCNQ2*, and *TBCID24*. Male pediatric patients had a significantly higher (57% vs. 29%; *p* < 0.05, odds ratio = 3.18) yield than their female counterparts. Seventeen genes were identified from the seizure groups, of which 82% were rare genetic etiologies for childhood seizure and did not appear recurrently in the case series. Conclusions: Wide genetic variation was identified for severe childhood seizures by WES. WES had a high yield, particularly in male infantile patients.

## 1. Introduction

In children, genetic disorders cause severe neurological disease, congenital malformation, inborn errors of metabolism, and developmental epileptic encephalopathy (DEE). DEEs refer to a group of ictal and interictal epileptiform anomalies (clinical and encephalographic) associated with severe cognitive and behavioral impairments according to the classification and terminology criteria of the International League against Epilepsy (ILAE) [1,2]. DEEs are age-specific and of diverse etiologies. Increasing evidence suggests that genetics play a pivotal role in pediatric DEEs and severe neurological disorders [3,4,5]. Although the incidence of each disease is low, the combined incidence is not adequately estimated and unknown. DEEs are highly heterogeneous genetically, but genetic etiologies have been identified in only half of the cases, typically in the form of de novo dominant mutations [6]. Diagnosing severe childhood neurological disorders is important but challenging.

DEEs are characterized by (a) early-onset seizures that are often intractable, (b) electroencephalographic abnormalities, (c) developmental delay or regression, and (d) early death in some patients [2,7,8]. In infants with *KCNQ2* or *STXBP1* encephalopathy, seizures may be controlled early after onset or may cease spontaneously after a few years. However, the developmental consequences are profound [3].

Despite the recent advances in molecular diagnostics, the genetic causes of only approximately 50% of patients have been identified [3,9]. The identification of the genetic etiology of DEE has greatly improved our understanding of disease pathophysiology at the molecular level. However, understanding the genotype–phenotype correlation remains a challenge. Some epileptic syndromes, such as Dravet syndrome (DS), are attributed to *SCN1A* mutations in approximately 80% of patients. Otahara syndrome is attributed to mutations in *KCNQ2, STXBP1*, and *ARX*. The outcomes are varied and may range from self-limited and drug-responsive to severe debilitating syndromes [9].

The aim of this study was to determine the diagnostic utility of whole-exome sequencing (WES) for a heterogeneous group of childhood DEEs and to develop a viable protocol for identifying severe neurological disease. We performed WES in 54 pediatric patients with suspected severe neurological or neuromuscular disorders, which remained etiologically undiagnosed despite comprehensive clinical, neuroimaging, and biochemical studies. Patients were enrolled from two medical centers in central Taiwan. The identified genetic etiologies are rare. The case series and diagnostic yield are discussed.

## 2. Materials and Methods

### 2.1. Patients

We enrolled patients from 2017 to 2020 that matched at least 1 of the following criteria: (1) Epileptic encephalopathy defined by ILAE, with the use of ≥2 AEDs, (2) epileptic seizures with the use of 1 AED but with symptoms of neurological regression, (3) nonepileptic patients with symptoms of neurological regression. We collected samples from 54 patients from 2 medical centers in central Taiwan, including the Chung Shan Medical University Hospital and the Children’s Hospital of China Medical University from 2017–2020. This study was approved by the institutional research ethics board (CMUH105-REC3-123) and Chung Shan Medical University Hospital’s Internal Review Board (IRB #: CS13036). Informed consent was obtained from the parents.

### 2.2. Extracting DNA from Peripheral Blood

A genomic DNA purification kit (Gentra Puregene Buccal Cell Kit; Qiagen Taiwan, Taipei City, Taiwan) was used to extract genomic DNA from peripheral blood. Three volumes of RBC lysis buffers were added to blood sample and mixed by inverting 30 times, incubated for 10 min at room temperature, and centrifuged 3000 rpm for 5 min. The supernatant was discarded. Next, we added 3 volumes of cell lysis buffers (3 mL) to the pellet and vortexed vigorously for 10 s to lyse the cell and centrifuged 3000 rpm for 5 min, followed by transferring the supernatant into a new 15 mL tube. Then, we added 3 mL isopropanol into a 15 mL tube, mixed by inverting 50 times, and centrifuged 3000 rpm for 5 min. The supernatant was carefully discarded and the pellet was air dried for 10 to 20 min. The dried pellet was resuspended in 300 μL nuclease-free water and frozen at −20 °C or −80 °C for storage. DNA were also extracted from their parents.

### 2.3. WES

We used the SureSelect XT HS2 DNA reagent kit (Agilent Technologies, Inc., Santa Clara, CA, USA) protocol for the Illumina Hiseq paired-end sequencing library (catalog# G9611A), and raw reads were mapped to the human genome assembly GRCh37 (also known as hg19) using Burrows-Wheeler Aligner software (version 0.6.1; Intel Corporation, Santa Clara, CA, United States). The SureSelectXT Human All Exon Version 6 (51 Mb) probe (Agilent Technologies, Inc., Santa Clara, CA, USA) set was used. For *Library* Preparation, 50 ng genomic DNA was used. The adapter-ligated *DNA* sample was purified using Agencourt Ampure Xp Pcr Purification Beads and analyzed using an Agilent DNA Kit. From the purified sample, the hybridization between DNA libraries (750 ng) and baits was carried out and purified using Agencourt Ampure Xp Pcr Purification Beads. The Agilent protocol was used for adding tags by Post-Hybridization Amplification. Finally, the sample was sequenced on Illumina NextSeq500 using the generated reads of 2 × 150 bp. Every analyte passed all the quality control requirements, and 99% of targeted nucleotides were covered at more than 20×.

### 2.4. Data Analysis and Interpretation

Variant calling was performed using the recommended best practices of GATK version1.0.5506 (Broad Institute). Variant annotation and prioritization were performed using a well-developed pipeline called wANNOVAR [10], which is used for functional annotations, including various gene annotations, alternative allele frequency in the 1000 Genomes Project, conserved element annotation, dbSNP annotation, deleteriousness prediction scores for nonsynonymous variants, ClinVar variant annotation, and genome-wide association study (GWAS) variant annotation [11].

We implemented a variant-reduction pipeline based on commonly used filters and disease models to select nonsynonymous variants and splice variants, rare or novel variants in the 1000 Genomes Project database, and predicted deleterious variants. Thus, synonymous variants, variants with variant frequency <10%, and variants with allele frequency >1% were removed. The remaining variants were annotated with reference to ClinVar, a freely accessible human variation and phenotype database hosted by the National Center for Biotechnology Information (NCBI) [12]. Pathogenicity prediction programs, including PolyPhen2 [13], SIFT [14], and Combined Annotation Dependent Depletion (CADD) [15], were used. All the variants identified were further confirmed by Sanger sequencing. The corresponding gene contexts were evaluated according to Online Mendelian Inheritance in Man^®^ [16] with the individual phenotypes (clinical, laboratory, and imaging data). Segregation analysis was performed to select de novo or compound heterozygous variants. The identified variants were classified “pathogenic,” “likely pathogenic,” or “of uncertain significance” according to the American College of Medical Genetics (ACMG) standards and guidelines [17].

## 3. Results

### 3.1. Demographic Data of the Enrolled Patients

A total of 54 patients were enrolled, of which 45 had a history of seizure, including 10 patients with seizure onset before 1 month of age, 31 patients with seizure onset between 1 month and 2 years of age, and 4 patients with seizure onset after 2 years of age. Of the patients, 10 received 1 drug and 35 received at least ≥2 AEDs. Nine patients had no history of seizures and were suspected to have severe neurological or neuromuscular disease with regression of symptoms. The demographic data are shown in Table 1. The detailed clinical information including antiepileptic drugs, magnetic resonance imaging (MRI) findings, and outcomes are demonstrated in Table 2.

### 3.2. Diagnostic Yield

The overall diagnostic rate was 44.4% (24/54). We compared patients with identified and nonidentified genetic etiologies through WES, which indicated a significant difference (*p* < 0.05) in sex between the two groups. In the identified genetic group (*N* = 24), the sex ratio was 18 males to 6 females. The inheritance patterns were 3 X-linked, 6 autosomal recessive, and 15 autosomal dominant. Male pediatric patients had a significantly higher yield than their female counterparts (57% vs. 29%, odds ratio = 3.18 (95% confidence interval = 1.04 to 9.70)). We found no difference in the number of epileptic drugs taken and the time of seizure onset between the identified genetic and nonidentified genetic groups (Table 1). Among 45 patients with seizures, the yield was 44% (20/45). Considerable genetic variability was found in the seizure group, as recurrent genetic etiologies included *KCNQ2, TBCID24*, and *SCN1A* in the cases series, with two patients carrying each gene. Other genes were nonrecurrent for each case in the case series. Seventeen genetic etiologies were identified in the seizure group, of which 82% (14/17) were rare and nonrecurrent.

### 3.3. Genotype

The clinical genotype and phenotype in genetically positive cases are shown in Table 2. Clinical and molecular data for 24 patients are listed. In 54 patients, 10 novel mutations were identified (Table 3). Among 24 patients with identified genetic etiologies, there were 13 with DEE, 6 patients had seizures with neurodevelopmental regressive symptoms, 2 patients with muscle disease (1 patient with muscle dystrophy carrying DMD gene and 1 patient with congenital muscle dystrophy with seizures), 1 with autism with *SCN2A* mutation, 1 with spinocerebellar ataxia, and 1 patient with Marfan syndrome. Eleven patients had a single autosomal-dominant disorder and all showed de novo variants. Of these, 10 mutations were novel (Table 3). The diagnosis in four patients was changed after WES, including one with migraine changed to spinocerebellar ataxia, one with Leigh-like syndrome changed to sulfite oxidase deficiency, and two which remained undiagnosed but were suspected of carrying copy number variation after read depth was computed from WES. Thereafter, a final diagnosis was given through multiplex ligation-dependent probe amplification (MLPA) or microarray-based comparative genomic hybridization (aCGH) (patients 9 and 17). The 10 novel mutations include 2 missense, 2 splicing, 3 nonsense, 2 indel, and 1 frameshift mutations. They were considered pathogenic or likely pathogenic mutations by ACMG guidelines (Table 3).

### 3.4. Refractory Seizure Cases

Of the 45 patients with seizure, a definitive genetic etiology was identified in 20 (44%). Of the 10 patients with seizure onset before the first month, etiology was confirmed through WES in 5 (50%) patients. These included two patients with *KCNQ2* and one patient each with *PACS1*, *OSGEP*, and *PIGA* (Table 2). All patients had poor neurodevelopmental outcomes. One patient with homozygous *OSGEP* mutations died before 4 months of age due to severe pleural effusions. One patient with *PACS1* has survived and is 15 years old.

Of the 31 patients with seizure onset between 1 month and 2 years of age, 13 (42%) received a definitive diagnosis, including 2 patients each with *TBCID24* and *SCN1A* and 1 patient each with *PPP1CB*, *UBE3A*, *MECP2* duplication syndrome (MECP2:rsa Xq28), *SZT2* (Figure 1), *SMARCA4* (Figure 2), *GRIN1*, [4p16.3p16.1(68,345–7,739,782)X1 17q25.1q25.3(73,608,322–81,041,938)X3], *SUOX*, and *LAMA2 SZT2* (Figure 3). Although the 13 patients were initially diagnosed with DEE, their phenotypes are heterogeneous.

A wide range of genetic disorders were identified. The genes involved in seizures included *KCNQ2, SCN1A, TBCID 24, GRIN1, IRF2BPL, MECP2, OSGEP, P*ACS1, *PIGA, PPP1CB, SMARCA4, SUOX*, SZT2, *UBE3A*, 16p13.11 microdeletion, [4p16.3p16.1(68,345–7,739,782)X1, 17q25.1q25.3(73,608,322–81,041,938)X3], and *LAMA2*. Most patients had poor neurological outcomes and refractory seizures controlled by AEDs.

### 3.5. Recurrent Mutations

The recurrent mutations involved *SCN1A, KCNQ2*, and *TBC1D24*. Six cases were attributed to these genes, with seizure onset before 2 years of age. The recurrence of these mutations in the case series indicates that these genes are more commonly involved in childhood DEE, particularly in cases where seizure onset occurs before 2 years of age. Two patients with mutations in *KCNQ2* had severe neurological outcomes, and they could not speak at 3 years of age or walk. Two patients with mutations in *TBCID24* had persistent refractory seizures with an attention deficit. Two patients with mutations in *SCN1A* had severe neurological outcomes and refractory seizures.

### 3.6. Genes Accounting for Epilepsy Demonstrated by the Age of Seizure Onset

The genes accounting for epileptic patients in the study and in a literature review [18,19,20,21,22,23] (Table 4) are demonstrated in the *order* of age of seizure onset from birth to 1 month old (Figure 4A), 1 month to 24 months old (Figure 4B), and 24 months to adulthood (Figure 4C).

### 3.7. Nonseizure Group

In the nonseizure group, the yield was 44% (4/9), including one with *SCN2A* with autism, one *FBN1* with Marfan syndrome, one *DMD* with muscle dystrophy, and one *SPTBN2* with spinocerebellar ataxia.

## 4. Discussion

Of 54 patients with severe neurological disorders, we identified variants corresponding with the disease in 24 patients. In neonatal infants with seizures, genetic etiologies varied. *SCN1A, KCNQ2*, and *TBCID24* had recurrent mutations in the case series. Mutations in other genes were sporadic and not recurrent. Seventeen genetic etiologies were identified in the seizure group. Of those, 82% (14/17) responsible for childhood seizure were rare and nonrecurrent, indicating that the genetic spectrum for neonatal infantile seizures is wide. Thus, the genes selected for the epileptic panel may not represent all etiologies.

The diagnostic yield of WES was 44%, which is not significantly different from that in other studies, which varied from 30% to 70% [19]. However, results vary greatly and depend on the disease groups selected. Studies have reported diagnostic rates of 19% [24], 41% [23], and 49.1% [25]. Neonatal or infantile DEE was the most commonly found syndrome in this study. Studies have used comprehensive epilepsy gene panel analysis, with a success rate ranging from 10% to 40% [20,21] depending on gene selection and number of cases. However, previous studies have not performed gene panel analysis, which explains why WES is more effective in diagnosing genetic etiologies.

The finding that the WES yield was significant in male pediatric patients provides a viable treatment strategy. The finding needs further explanation: One factor is X-linked inheritance. The sex ratio was 18 males to 6 females. However, only 3 were X-linked. The second factor is that the majority (76%) of 54 patients were aged below 2 years old. In the identified genetic group, seizures onset before 1 year old were found in 18 (75%) patients. Demos et al. [18] found that 12 (44.4%) of 27 males and 9 (35%) of 26 females with epileptic diagnosis <6 months had a genetic diagnosis by WES in 360 epileptic cases with an average seizure onset at ≤5 years (Table 4). That indicated a trend of higher yield in male newborn infants with a younger age of seizure onset. Genetic counseling to prevent next offspring was carried out in all patients, even in the autosomal dominant patients with de novo mutation. In our cohort, 12 patients with autosomal dominant inheritance had de novo mutations. The next offspring were explained to prevent unnecessary concern. In addition, we changed antiepileptic drug selection after detecting a causative *KCNQ2* variant in patients 4 and 5, which resulted in more effective seizure control. A patient was identified with a mutation in *KCNMA1*, which encodes the α-subunit of the large conductance calcium-sensitive potassium channel. Mutations in *KCNMA1* increase Ca^2+^ sensitivity of the channel by three- to five-fold, resulting in generalized epilepsy and paroxysmal dyskinesia [24]. We chose to administer levetiracetam (LEV; 20 mg/kg/day) because LEV can limit epileptogenesis by inhibiting Ca^2+^ elevation following seizures, thus exerting neuroprotective and antiepileptogenic effects [26,27]. The administration of LEV resulted in no episodes of seizure and low frequency of paroxysmal dyskinesia in the patient. Thus, WES data can support precision therapy and provide information for managing pediatric patients with epilepsy.

*SCN1A* is studied in the context of genotype–phenotype correlations in the GEFS+ spectrum and DS [20,28,29]. Pathological mutations in structural and functional proteins correlate with specific clinical presentations. Modifier genes also contribute to modulating the phenotype. Most phenotypes that cause severe disability are a result of de novo mutations [20,30]. However, the hypothesis does not account for mosaicism [31]. Some studies have proposed that epigenetic factors contribute to determining phenotypes [20,30,31]. In *SCN1A* channelopathy, DS is the most severe phenotype. Approximately 70–80% of patients with DS [20,30] have *SCN1A* mutations, which are mostly de novo, as in our patients. Most patients with DS have mutations in the “ion pore” of SCN1A [32,33,34,35]. However, one of our patients with a mutation in the ion pore region had an identical unaffected twin sister. Although studies have reported that patients with pathogenic *SCN1A* mutations may remain unaffected, the mechanism is unknown. In a large family with familial pathogenic *SCN1A* missense mutations, clustering of three unaffected carriers was observed in the same generation [35]. Genetic or epigenetic changes were proposed as the causative factors but not established, because the effects of other confounding factors could not be excluded. *KCNQ2* encodes a voltage-gated potassium channel that is expressed in the brain and is involved in the etiology of epileptic encephalopathy, early infantile (EIEE7, phenotype MIM# 613720), and benign familial neonatal seizures-1 (BFNS1, phenotype MIM#121200) [36,37]. Two (patients 4 and 5) de novo heterozygous mutations in *KCNQ2*, namely c.740C > T (p.Ser247Leu) and c.740C > T (p.Pro285Thr), are highly pathogenic and located in the critical pore domain [38]. This finding helps us select suitable AEDs, such as oxcarbazepine, to control seizures.

The diagnostic methods focusing on the genotype–phenotype correlation followed by specialized biochemical tests and Sanger sequencing for suspected genes are time-consuming and costly [39]. Moreover, only “typical” diseases can be diagnosed, and even when the same causative mutations are present, atypical diseases can be overlooked. In summary, we provide a perspective for diagnosing severe neurological disorders in children using WES. The study presents important findings on the genetic causes of severe neurological disorders in childhood, particularly in epilepsy and neurodegenerative diseases.

In the clinical setting, if the clinical course is thought to be benign, including benign epileptic syndromes [40] in childhood, WES is not performed. However, if, after treatment with AEDs, the response is unfavorable and seizures are refractory, to obtain the best time of seizure control and to avoid sacrificing the moment of children’s brain development, as in our case with *UBE3A* mutation whose seizure was far preceded to her developmental regression, WES should be performed. Therefore, we recommend WES for assessing all childhood seizures with poor response to AEDs without definitive etiology to save time in selecting appropriate AEDs.

## 5. Conclusions

The findings on the genetic causes of severe childhood neurological disorders are significant. A scientific and efficient WES analysis is particularly important and should be emphasized because the affected gene may not be present in the panel. This findings of this study highlight the diagnostic relevance of WES for childhood patients with epilepsy in clinical practice.

## Figures and Tables

**Figure 1 jpm-10-00281-f001:**
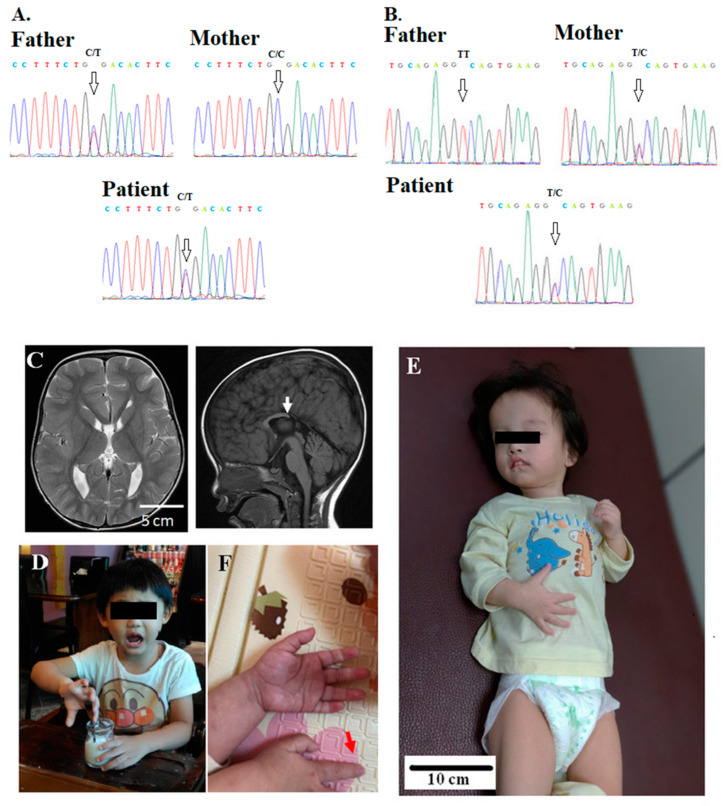
Patient 21 has compound heterozygous mutations of *SZT2*, (**A**) c.C9055T (p.Arg3019Ter) (arrows) and (**B**) c.1496 + 2T > C (arrows), from the asymptomatic father and mother, respectively. Seizure onset occurred at 6 months old. His electroencephalogram shows developmental epileptic encephalopathy with multifocal epileptiform discharges. (**C**) The MRI shows a short and thick corpus callosum with missing of selenium (arrow). (**D**) Normal child with the same age demonstrates for comparison with patient. (**E**) Her examination was notable for macrocephaly, dysmorphic features, frontal bossing, hypertelorism, microphthalmia, depressed nasal bridge, long-tapered fingers, and hyperextensible joints. (**F**) Long-tapered fingers in the patient (red arrow).

**Figure 2 jpm-10-00281-f002:**
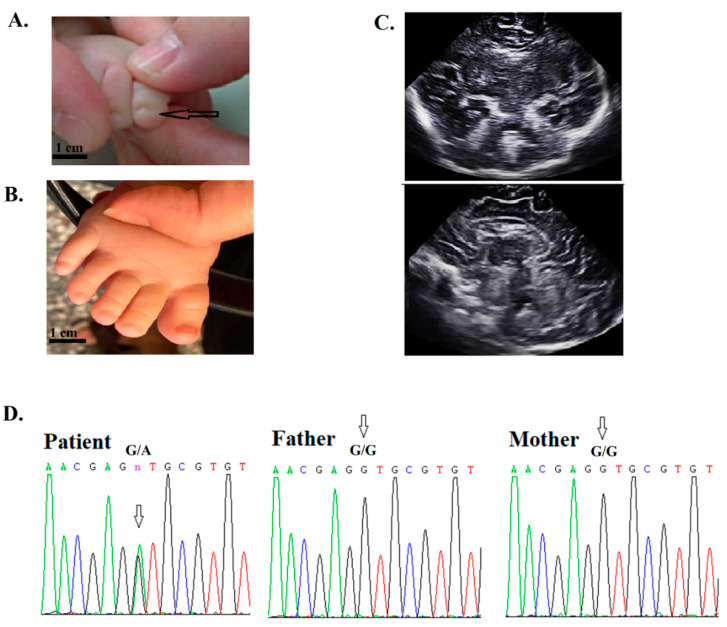
Patient 3 had facial dysmorphic feature, hypotonia, and orogastric tube dependency after birth. (**A**) His toenails were hypoplastic (arrow). Seizure onset occurred at 3 months old. (**B**) Normal toenails in a healthy infant. (**C**) The image of the head ultrasound was unremarkable. (**D**) The WES and sequencing results indicated a novel de novo (no mutation in both parents) mutation in the *SMARCA4* gene. The arrow indicates a G-to-A substitution at nucleotide 3595 (c.3595G > A, p.Val1199Met) in the patient (arrow), father, and mother. At 2 years, the patient was not able to sit without support.

**Figure 3 jpm-10-00281-f003:**
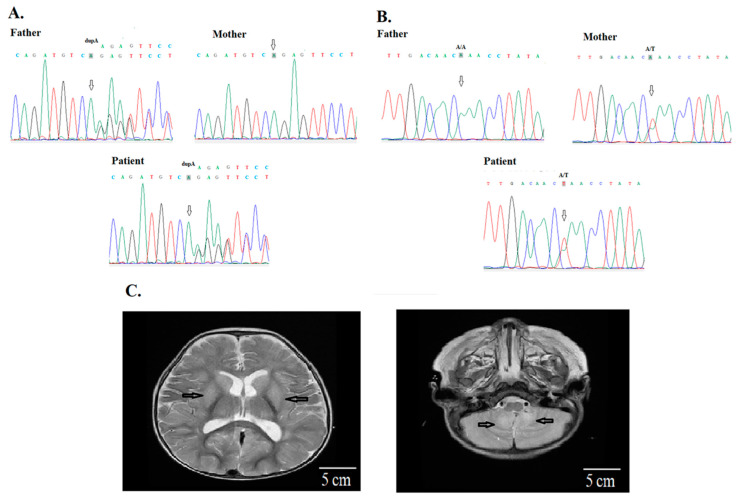
Patient 24 has compound heterozygous mutations of *LAMA2*, (**A**) c.1583dupA (p.Ser529GlufsTer19), and (**B**) c.6931A > T (p.Lys2311Ter) from the asymptomatic mother and father, respectively (arrows). (**C**) His MRI showed basal ganglion, thalamus, white matter and cerebellar vermis signal abnormalities. The finding indicated a dysmyelinating process. A creatine phosphokinase was 548 IU/L (reference values, 40-397 IU/L). The genetic study for spinal muscle atrophy was unremarkable. He had developmental delay, recurrent sepsis, and pneumonia with lung atelectasis, and needed respiratory supportive care at home since birth. He died at 1 year and 2 months old due to respiratory failure.

**Figure 4 jpm-10-00281-f004:**
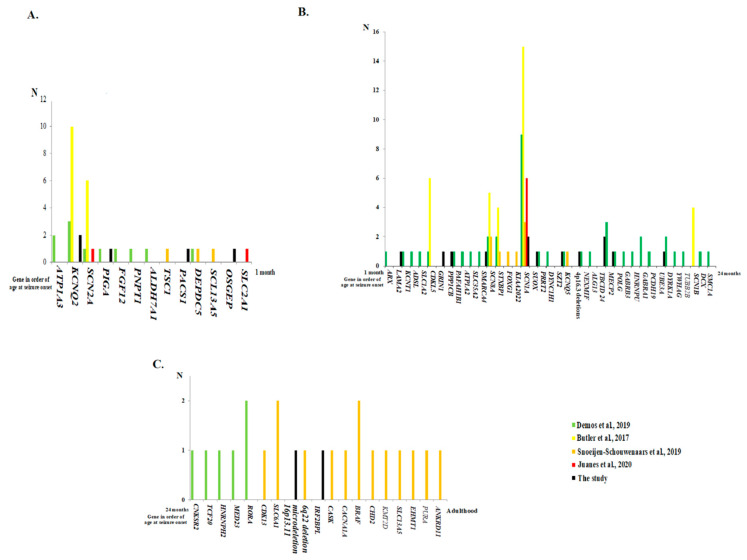
The genes accounting for epilepsy are demonstrated in the *order* of age of seizure onset from birth to 1 month old (**A**), from 1 month to 24 months old (**B**), and from 24 months to adulthood (**C**) from the study and from a literature review.

**Table 1 jpm-10-00281-t001:** The demographic data in identified genetic group and nonidentified genetic group by whole-exome sequencing (WES) are demonstrated.

Clinical Data (*N*)	Identified Genetic Group (*N* = 24)	Nonidentified Genetic Group (*N* = 30)	
**Symptoms**			NS
**Seizures (45)**	20 (44%)	25 (56%)	
**No seizure (9)**	4 (44%)	5 (56%)	
**Age at first seizure onset**			NS
**0–1 month (10)**	5 (50%)	5 (50%)	
**1 month–2 years (31)**	13 (42%)	18 (58%)	
**>2 years (4)**	2 (50%)	2 (50%)	
**No seizure (8)**	4 (44%)	5 (56%)	
**Sex**			*p* < 0.05 ^※^
**Male (30)**	17 (57%)	13 (43%)	
**Female (24)**	7 (29%)	17 (71%)	
**Number of antiepileptic drugs at the time of study**			NS
**0 (8)**	4 (44%)	5 (56%)	
**1 (10)**	2 (20%)	8 (80%)	
**2 (11)**	5 (49%)	6 (51%)	
**>3 or =3 (25)**	13 (54%)	11 (46%)	

NS indicates not significant. **^※^** Odds ratio = 3.18 (95% confidence interval = 1.04 to 9.70).

**Table 2 jpm-10-00281-t002:** The genetic and clinical features are summarized in those patients with identified mutations by WES.

	Gene Name	Genotype	Inheritance Pattern	Phenotype	Age of Diagnosis/ Age of First Seizure	Sex	NCBI ClinVar	Seizure Types	Neurodevelopmental Outcomes
**P 1**	*PACS1* (NM_018026.4)	c.607C > T p.(Arg203Trp)	AD, *de novo*	DEE, spinal tethered cord	12 years/ Newborn	F	Pathogenic	Focal seizures	Walk by ambulance, cognition delay at 14 years
**P 2**	*TBCID 24* (NM_001199 107.1)	c.1499C > T (p.Ala500Val)/c.229_240del (p.77_80del)	AR	DEE	2 years /10 months	M	Pathogenic/Pathogenic	Multiple focal, general, nonconvulsive status epilepticus	Attention deficit. cognition delay
**P 3**	*FBN1* (NM_000138. 4)	c.794_795insGTAT (p.Val266Tyr fs * 6)	AD	Marfan syndrome	16 years /-	F	Pathogenic	No seizure	Normal
**P 4**	*KCNQ2* (NM_172107.3)	c.740C > T (p.Ser247Leu)	AD, *de novo*	DEE	10 months /Day 3	M	Pathogenic	General tonic, apnea	No language, hypotonia at 3 years
**P 5**	*KCNQ2* (NM_172107.3)	c.853C > A (p.Pro285Thr)	AD, *de novo*	DEE	2 months /Day 3	F	Novel	General tonic	No language, hypotonia at 1 year
**P 6**	*OSGEP* (NM_017807.4)	c.740G > A (p.Arg247Gln)/ c.740G > A (p.Arg247Gln)	AR	Lissencephaly, seizures	3 months /3 weeks	M	Pathogenic/Pathogenic	Focal	Severe, died at 6 months
**P 7**	*SCN2A* (NM_021007.2)	c.4958delT (p.Leu1653Ter)	AD, *de novo*	Autism	4 years/-	M	Pathogenic	No seizure	Severe autism, no language at 4 years
**P 8**	16p13.11 microdeletion		AD	Frequent seizures	12 years/10 years	F	Pathogenic	General tonic	Attention deficit
**P 9 ^※^**	4p16.3p16.1(68,345–7,739,782)X1, 17q25.1q25.3(73,608,322–81,041,938)X3		AD, *de novo*	DEE	1 year/ 6 months	M	Pathogenic	General	Global DD
**P 10**	*IRF2BPL* (NM_024496.3)	c.562C > T (p.Arg188Ter)	AD, de novo	DEE and regressive encephalopathy	12 years /11 years	F	Pathogenic	General	Profound ID
**P 11**	*PIGA:* (NM.002641)	c.356G > A (p.Arg119Gln)	X-linked	DEE	0.5 month /Day 3	M	Pathogenic	Apnea, cyanosis, absence, atonic, spasms	Severe global DD
**P 12**	*PPP1CB*	c.548A > C (p.Glu183Ala)	AD, de novo	DEE, dysmorphism	4 months/ 2 months	M	Pathogenic	Apnea, eye gazed deviation, myoclonic	Severe global DD
**P 13**	*TBCID 24* (NM_001199 107.1)	c.119G > A (p.Arg40His)/ c.1499C > T (p.Ala500Val)	AR	DEE	8 months /6 months	M	Pathogenic /Pathogenic	Multifocal, myoclonus	Global DD
**P 14**	*UBE3A* (NM_130838)	c.C219A (p.Thr73Ter)	AD, maternal imprinting	Regressive encephalopathy	14 months /11 months	M	Novel	General tonic	Profound ID
**P 15**	*SCN1A* (NM_001165963)	c.362delC (p.Ala121fs)	AD, de novo	DEE	5 months /4 months	M	Novel	Focal clonic, general tonic	Profound ID
**P 16**	*SCN1A* (NM_001165963)	c.3918 + 1 G > -	AD, de novo	DEE	8 months /5 months	F	Novel	Focal clonic, general tonic	Profound ID
**P 17 ^$^**	*MECP2* (NM_004992.3)	rsa Xq28 MECP2 (exons3-4) x1	X-linked	Regressive encephalopathy	14 months /8 months	F	Pathogenic	General tonic	Profound ID
**P 18**	*DMD* (NM_004006.2)	c.C5287T (p.Arg1763Ter)	X-linked	Muscle dystrophy	3 years /-	M	Pathogenic	No seizure	Motor delay
**P 19**	*SMARCA4* (NM_001128849)	c.3595G > A (p.Val1199Met)	AD, de novo	Seizures, encephalopathy	3 months /2 months	M	Novel	General tonic	Severe global DD
**P 20**	*GRIN1* (NM_007327.3)	c.C2414T (p.Pro805Leu)	AD, de novo	DEE	4 months /2 months	M	Pathogenic	Infantile spasms	Global DD
**P 21**	*SZT2* (NM_015284)	c.1496 + 2T > C/c.9055T > C (p.Arg3019Ter)	AR	DEE	3 years /6 months	M	Novel/Novel	Focal motor	Severe hypotonia, gastrostomy
**P 22**	*SUOX* (NM_001032386.2)	c.650G > A (p.Arg217Gln)/ c.258dupT (p.Lys87Ter)	AR	Leigh-like, regression	8 months /4 months	M	Pathogenic /novel	Apnea, cyanosis, *opisthotonos*	Severe DD, hypertonia
**P 23**	*SPTBN2* (NM_006946. 2)	c.5515C > A (p.Gln1839Lys)	AD	Ataxia, vertigo	14 years /-	M	Novel	No seizure	Frequent ataxia
**P 24**	*LAMA2* NM_000426.3	c.1583dupA (p.Ser529GlufsTer19) /c.6931A > T (p.Lys2311Ter)	AR	Hypotonia, seizures	2 years /1.5 months	M	Novel/Novel	Focal	Severe global DD

**^※^** Diagnosed by WES and 750K microarray chip (Affymetrix CytoScan 750K Array); ^$^ Diagnosed by WES and multiplex ligation-dependent probe amplification (MLPA). P, patient; DEE, developmental epileptic encephalopathy; F, female; M, male; AD, autosomal dominant; AR, autosomal recessive; DEE, developmental epileptic encephalopathy; DD, developmental delay; ID, intelligence disability; NA, not available; NCBI ClinVar, National Center for Biotechnology Information, clinical variability and predictability [12]. The sequence data of each patient were checked against the GenBank reference sequence and version number of genes.

**Table 3 jpm-10-00281-t003:** Ten novel mutations in the cases-series.

	Gene Name	Genotype (*N*)		Phenotype	NCBI ClinVar	Critical Functional	Global	East Asia	Taiwan Biobank.	CADD	Poly	SIFT	ACMG
						Domain	MAF	MAF		Predict	Phen2		Score
**Patient 15**	*SCN1A* (NM_001165963)	c.362delC (p.Ala121fs)	Frameshift	DEE	Novel		0	0	0	D	D	D	AD, de novo, (PVS1, PM2, PP3)
**Patient 14**	*UBE3A* (NM_130838)	c.C219A (p.Thr73Ter)	Nonsense	Encephalopathy and regression	Novel		0	0	0	D	D	D	AD from mother (PVS1, PM2, PP3)
**Patient 16**	*SCN1A* NM_001165963)	c.3918 + 1 G > -	Splicing	DEE	Novel		0	0	0	D	D	D	AD, de novo (PVS1, PM2, PP3)
**Patient 19**	*SMARCA4* (NM_001128849)	c.3595G > A (p.Val1199Met)	Missense	Seizure, severe global delay	Novel	Helicase C-terminal’	0	0	0	D	D	D	AD, de novo (PS2, PM1, PM2, PP3)
**Patient 21**	*SZT2* (NM_015284)	c.1496 + 2T > C	Splicing	DEE	Novel		0	0	0	D	D	D	AR (PVS1, PM2, PP3)
**Patient 21**	*SZT2* (NM_015284)	c.9055T > C (p.Arg3019Ter)	Nonsense	DEE	Novel		0	0	0	D	D	D	AR (PVS1, PM2, PP3)
**Patient 22**	*SUOX* (NM_001032386.2)	c.258dupT (p.Lys87Ter)	Indel	Leigh-like, regression	Novel		0	0	0	D	D	D	AR (PVS1, PM2, PP3)
**Patient 23**	*SPTBN2* (NM_006946. 2)	c.5515C > A (p.Gln1839Lys)	Missense	Spinocerebellar ataxia 5	Novel	Ankyrin binding domain	0	0	0	D	D	T	AD from mother (PM1 + PM2 + PP1 + PP3 + PP4)
**Patient 24**	*LAMA2* NM_000426.3	c.1583dupA(p.Ser529GlufsTer19)	Indel	CMD, seizures	Novel		0	0	0	D	D	D	AR (PVS1, PM2, PP3)
**Patient 24**	*LAMA2* NM_000426.3	/c.6931A > T (p.Lys2311Ter)	Nonsense	CMD, seizures	Novel		0	0	0	D	D	D	AR (PVS1, PM2, PP3)

DEE, developmental epileptic encephalopathy; CMD, Congenital muscular dystrophies; NCBI ClinVar, National Center for Biotechnology Information, clinical variability and predictability [12]; Global MAF, Global mutation allele frequency in EXAC browser; East Asia MAF, East Asia mutation allele frequency in EXAC browser; CADD predict, Combined Annotation Dependent Depletion prediction; D, damage; T, tolerant. ACMG, American College of Medical Genetics and Genomics and the Association for Molecular Pathology. The sequence data of each patient were checked against the GenBank reference sequence and version number of genes.

**Table 4 jpm-10-00281-t004:** Diagnostic yield from different methods to identify genetic causes.

Reference	The Study	[18]	[19]	[20]	[21]	[22]	[23]
**Method**	WES	WES	WES	110-gene panel	47-gene panel	WGS	WES
**Cases (*N*)**	54	360	100	339	17	14	78
**Ethnicity or region**	Taiwan	Canada	Caucasian (2 patients had African ethnicity), Netherlands	United States	Argentinean	United States	United States
**Age**	43.7 ± 62.0 months 41 (76%) seizure onset <2 years	Seizure onset was 18 months (range 0.03–60 months)	24.1 ± 16.2 years (ranged 2.8–67.6 years)	Age ranged 2.5 months to 74 years	Average age 4 months (range: 0–10 months)	Seizure onset before first month	8.6 ± 5.8 years (ranged 1.6–26.3 years)
**Inclusion criteria**	45 epilepsy (35 with AEDs ≥2; 10 with 1 AED and regression) + 9 nonepileptic with regressive symptoms	Seizure onset at ≤5 years	Unexplained epilepsy and borderline intelligence disability	Epileptic patients	EE with age of onset under 12 months	EIEE	Neurodevelopmental disabilities
**Male/Female**	30/24	154/206	55/45	About equal	8/9	5/9	41/37
**Yield**	44%	33%	25%	18%	∼50%	100%	41%
**Male/Female in identified** **genetic group**	17 (57%)/ 7 (29%)	23/154 (15%)/ 36/206 (19%); In 53 epilepsy diagnosis < 6 months, 12/27 (44.4%) in males and 9/26 (35%) in females	12 (22%)/ 13 (29%)	NA	4 (50%)/ 4 (44%)	5 (100%)/ 9 (100%)	NA

WES, whole-exome sequencing; WGS, whole-genome sequencing; EE, epileptic encephalopathy; AED, antiepileptic drug; NA, not available from the reference.

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
