# Peer review of "A Wide Spectrum of Genetic Disorders Causing Severe Childhood Epilepsy in Taiwan: A Case Series of Ultrarare Genetic Cause and Novel Mutation Analysis in a Pilot Study"

_jpm, 2020, doi:10.3390/jpm10040281_

Round 1
Reviewer 1 Report
The authors analyzed a relatively large number of children with epilepsy, looking for the genetic disorders. They used WES (while-exome sequencing) from blood samples. Genetic diagnoses were confirmed in 24 (out of 57) patients. The study is sound and important, because the genetic diagnosis helps the choice of the appropriate pharmacotherapy. I have a very few suggestions:
- Fig.1. displays a corpus callosum dysgenesis - it seems that the selenium is missing. It should be noted in the Fig. caption.
- Fig.2. the ultrasound pictures are of bad quality.
- Fig.3. the MRI picture displays sose abnormalities which should be precisely stated in the Fig. caption (microgyria?, myelination problem?, tumor?).
Author Response
December 6, 2020
Dear Editor-in-Chief and Reviewers:
Manuscript ID: jpm-1029241, Type of manuscript: Article
We are grateful for the opportunity to improve our manuscript and we thank the editorial board and the reviewers for their thoughtful and helpful comments and criticisms. We have modified the paper as suggested. Following are our point-by-point responses. We have also highlighted the principal changes in the “manuscript with the changes marked”
We hope that you, the editorial board, and the reviewers now find our manuscript appropriate for publication in Journal of Personalized Medicine.
We look forward to your reply.
Sincerely yours,
Inn-Chi Lee, MD, PhD
Corresponding Author
The authors analyzed a relatively large number of children with epilepsy, looking for the genetic disorders. They used WES (while-exome sequencing) from blood samples. Genetic diagnoses were confirmed in 24 (out of 57) patients. The study is sound and important, because the genetic diagnosis helps the choice of the appropriate pharmacotherapy. I have a very few suggestions:
- 1. displays a corpus callosum dysgenesis - it seems that the selenium is missing. It should be noted in the Fig. caption.
Reply: Thanks for the valuable opinions. We have changed the figure legend according to reviewers’ suggestions.
In Figure 1, legend:
“Figure 1. Patient 21 has compound heterozygous mutations of SZT2, (A) c.C9055T (p.Arg3019Ter) (arrows) and (B) c.1496+2T>C (arrows), from the asymptomatic mother and father respectively. Seizure onset occurred at 6 months old. His electroencehalogram shows developmental epileptic encephalopathy with multifocal epileptiform discharges. (C) The MRI shows a short and thick corpus callosum (arrow). (D) Her examination was notable for macrocephaly, dysmorphic features, frontal bossing, hypertelorism, microophthalmia, depressed nasal bridge, long-tapered fingers, and hyper-extensible joints.”
Change to:
“Figure 1. Patient 21 has compound heterozygous mutations of SZT2, (A) c.C9055T (p.Arg3019Ter) (arrows) and (B) c.1496+2T>C (arrows), from the asymptomatic mother and father respectively. Seizure onset occurred at 6 months old. His electroencehalogram shows developmental epileptic encephalopathy with multifocal epileptiform discharges. (C) The MRI shows a short and thick corpus callosum with missing of selenium (arrow). (D) Normal child with the same age demonstrates for comparison with patient. (E) Her examination was notable for macrocephaly, dysmorphic features, frontal bossing, hypertelorism, microophthalmia, depressed nasal bridge, long-tapered fingers, and hyper-extensible joints. (F) long-tapered fingers in the patient (red arrow).”
- 2. the ultrasound pictures are of bad quality.
Reply: We have redone Figure 2 according the reviewers’ suggestions. The de novo mutation mean no mutation found in both parents.
“Figure 2. Patient 3 had facial dysmorphic feature, hypotonia, and orogastric tube dependency after birth. (A) His toenails were hypoplastic. Seizure onset occurred at 3 months old. (B) The image of head ultrasound was unremarkable. (C) The WES and sequencing results indicated a novel de novo mutation in the SMARCA4 gene. The arrow indicates a G-to-A substitution at nucleotide 3595 (c.3595G>A, p.Val1199Met) in the patient (arrow), father, and mother. At 2 years, the patient was not able to sit without support.”
Change to:
“Figure 2. Patient 3 had facial dysmorphic feature, hypotonia, and orogastric tube dependency after birth. (A) His toenails were hypoplastic (arrow). Seizure onset occurred at 3 months old. (B) Normal toenails in a healthy infant. (C) The image of head ultrasound was unremarkable. (D) The WES and sequencing results indicated a novel de novo (no mutation in both parents) mutation in the SMARCA4 gene. The arrow indicates a G-to-A substitution at nucleotide 3595 (c.3595G>A, p.Val1199Met) in the patient (arrow), father, and mother. At 2 years, the patient was not able to sit without support.”
- 3. the MRI picture displays sose abnormalities which should be precisely stated in the Fig. caption (microgyria?, myelination problem?, tumor?).
Reply: Thanks for the valuable opinion. The MRI indicates dysmyelination. We have redone the Figure 3 and changed the legend accordingly.
Figure 3. Patient 24 has compound heterozygous mutations of LAMA2, (A) c.1583dupA (p.Ser529GlufsTer19) and (B) c.6931A>T (p.Lys2311Ter), from the asymptomatic mother and father respectively (arrows). (C) His MRI showed basal ganglion, thalamus, white matter and cerebellar vermis signal abnormalities. A creatine phosphokinase was 548 IU/L (reference values, 40-397 IU/L). The genetic study for spinal muscle atrophy was unremarkable. He had developmental delay, recurrent sepsis, and pneumonia with lung atelectasis, and needed respiratory supportive care at home since birth. He died at 1years and 2 months old due to respiratory failure.
Changed to:
“Figure 3. Patient 24 has compound heterozygous mutations of LAMA2, (A) c.1583dupA (p.Ser529GlufsTer19) and (B) c.6931A>T (p.Lys2311Ter), from the asymptomatic mother and father respectively (arrows). (C) His MRI showed basal ganglion, thalamus, white matter and cerebellar vermis signal abnormalities. The finding indicated dysmyelinating process. A creatine phosphokinase was 548 IU/L (reference values, 40-397 IU/L). The genetic study for spinal muscle atrophy was unremarkable. He had developmental delay, recurrent sepsis, and pneumonia with lung atelectasis, and needed respiratory supportive care at home since birth. He died at 1 year and 2 months old due to respiratory failure.”
Comments and Suggestions for Authors in Reviewer 2:
- The topic of this manuscript is interesting and valuable. The overall case number was relatively small, so it is a pioneer study which should be indicated in the title. The manuscript provides sufficient background and proper description of the methods. However, the tables and the supplementary were not provided (I cannot find them from the reviewer page) and it is difficult to peer review it without these results. The quality of the figures was not good enough to be published, but these can be modified. I suggest considering the manuscript after a major revision.
Reply:
- Thanks for the valuable opinion. We have provided the Table 1-3 and Table 4 (old Supplementary Table 1) in the “Text file”
- We changed the “Supplementary Table 1” to “Table 4”.
- We have redone the Figure 1, 2 and 3 according to reviewers’ suggestions.
- In the Title:
We change the title:
“A Wide Spectrum of Genetic Disorders Causing Severe Childhood Epilepsy in Taiwan: A Case-Series of Ultrarare genetic cause and Novel Mutation Analysis” to
“A Wide Spectrum of Genetic Disorders Causing Severe Childhood Epilepsy in Taiwan: A Case-Series of Ultrarare genetic cause and Novel Mutation Analysis in a Pilot Study”
- In line 292, we deleted “Supplementary Materials: The following are available online. Table S1: Diagnostic yield from different methods to identify genetic causes”
Here are some minor points for the modification:
- Line 42 please provide exact incidence of the disease.
In line 42:
“Although the incidence of each disease is low, the combined incidence is high.”
Change to:
“Although the incidence of each disease is low, the combined incidence is not adequately
estimated and unknown.”
- Line 79 Please provide the details of the Primers for PCR amplification.
Reply: Thanks for the opinion. This method is to describe the extracting DNA from Peripheral Blood. Because of this, no primers for PCR were necessary.
In Method:
2.2. Extracting and Amplifying DNA from Peripheral Blood Using PCR
A genomic DNA purification kit (Gentra Puregene Buccal Cell Kit; Qiagen Taiwan, Taipei City) was used to extract genomic DNA from peripheral blood. Briefly, 100 ng genomic DNA was mixed with 10 mM Tris-HCl (pH 9.0), 1.5 mM MgCl2, 50 mM KCl, 0.1% (w/v) gelatin, 1% Triton X-100, 0.2 mM dNTPs, 0.5 mM primers, and 1 unit Taq DNA polymerase (ProTech Technology Enterprise Co., Taipei, Taiwan). The PCR reaction included 35 cycles of 30 s at 94°C, annealing at a special temperature for 30 s, and extension at 72°C for 1 min.
Change to :
“2.2. Extracting DNA from Peripheral Blood
A genomic DNA purification kit (Gentra Puregene Buccal Cell Kit; Qiagen Taiwan, Taipei City) was used to extract genomic DNA from peripheral blood. DNA for WES was extracted from index patients and their parents and was stored at −80°C.”
- Line 98 subtitle number was missing
Reply: We added the subtitle “2.4.”
- line 125 All the tables are missing
Reply: The “Supplementary Table 1” has been changed to “Table 4”
We have provided the Table 1-4 in the Text file.
- line 169 A, the letters are too small and not clear. It is recommended to show align figures for all of the sequencing comparison. CD, please make sure the pictures are not deformed. Please also show the scale bar.
Reply: Thanks for the opinion. We have redone the Figure 1 according to the suggestions.
- line 174 the picture failed to show the feature very well. It is recommended to show local figures to obviously demonstrate the abnormity. Figures from normal children are also recommended to be shown beside the patients’ figures for the comparison.
Reply: We have redone the Figure 1 according to these suggestions.
In Figure 1, legend:
“Figure 1. Patient 21 has compound heterozygous mutations of SZT2, (A) c.C9055T (p.Arg3019Ter) (arrows) and (B) c.1496+2T>C (arrows), from the asymptomatic mother and father respectively. Seizure onset occurred at 6 months old. His electroencehalogram shows developmental epileptic encephalopathy with multifocal epileptiform discharges. (C) The MRI shows a short and thick corpus callosum (arrow). (D) Her examination was notable for macrocephaly, dysmorphic features, frontal bossing, hypertelorism, microophthalmia, depressed nasal bridge, long-tapered fingers, and hyper-extensible joints.”
Change to:
“Figure 1. Patient 21 has compound heterozygous mutations of SZT2, (A) c.C9055T (p.Arg3019Ter) (arrows) and (B) c.1496+2T>C (arrows), from the asymptomatic mother and father respectively. Seizure onset occurred at 6 months old. His electroencehalogram shows developmental epileptic encephalopathy with multifocal epileptiform discharges. (C) The MRI shows a short and thick corpus callosum with missing of selenium (arrow). (D) Normal child with the same age demonstrates for comparison with patient. (E) Her examination was notable for macrocephaly, dysmorphic features, frontal bossing, hypertelorism, microophthalmia, depressed nasal bridge, long-tapered fingers, and hyper-extensible joints. (F) long-tapered fingers in the patient (red arrow).”
- line 177 A, please enlarge the toenails to be clearer. Figures from normal children are also recommended to be shown beside the patients’ figures for the comparison. C, please point out the corresponding base pair in parental sequences.
Reply: Thanks for the valuable opinions. We have redone Figure 2 to be more clear accordingly. The de novo mutation mean no mutation found in both parents.
In Figure 2, legend:
“Figure 2. Patient 3 had facial dysmorphic feature, hypotonia, and orogastric tube dependency after birth. (A) His toenails were hypoplastic. Seizure onset occurred at 3 months old. (B) The image of head ultrasound was unremarkable. (C) The WES and sequencing results indicated a novel de novo mutation in the SMARCA4 gene. The arrow indicates a G-to-A substitution at nucleotide 3595 (c.3595G>A, p.Val1199Met) in the patient (arrow), father, and mother. At 2 years, the patient was not able to sit without support.”
Change to:
“Figure 2. Patient 3 had facial dysmorphic feature, hypotonia, and orogastric tube dependency after birth. (A) His toenails were hypoplastic (arrow). Seizure onset occurred at 3 months old. (B) Normal toenails in a healthy infant. (C) The image of head ultrasound was unremarkable. (D) The WES and sequencing results indicated a novel de novo (no mutation in both parents) mutation in the SMARCA4 gene. The arrow indicates a G-to-A substitution at nucleotide 3595 (c.3595G>A, p.Val1199Met) in the patient (arrow), father, and mother. At 2 years, the patient was not able to sit without support.”
- line 184 A, the letters are too small and not clear. some figures are not complete (The signal of negative bp were cut out). C, please make sure the pictures are not deformed. Please also show the scale bar.
Reply: We have redone the Figure 3 and change the legend accordingly.
“Figure 3. Patient 24 has compound heterozygous mutations of LAMA2, (A) c.1583dupA (p.Ser529GlufsTer19) and (B) c.6931A>T (p.Lys2311Ter), from the asymptomatic mother and father respectively (arrows). (C) His MRI showed basal ganglion, thalamus, white matter and cerebellar vermis signal abnormalities. A creatine phosphokinase was 548 IU/L (reference values, 40-397 IU/L). The genetic study for spinal muscle atrophy was unremarkable. He had developmental delay, recurrent sepsis, and pneumonia with lung atelectasis, and needed respiratory supportive care at home since birth. He died at 1years and 2 months old due to respiratory failure.”
Change to:
“Figure 3. Patient 24 has compound heterozygous mutations of LAMA2, (A) c.1583dupA (p.Ser529GlufsTer19) and (B) c.6931A>T (p.Lys2311Ter), from the asymptomatic mother and father respectively (arrows). (C) His MRI showed basal ganglion, thalamus, white matter and cerebellar vermis signal abnormalities. The finding indicated dysmyelinating process. A creatine phosphokinase was 548 IU/L (reference values, 40-397 IU/L). The genetic study for spinal muscle atrophy was unremarkable. He had developmental delay, recurrent sepsis, and pneumonia with lung atelectasis, and needed respiratory supportive care at home since birth. He died at 1 year and 2 months old due to respiratory failure.”
- line 188 size of the “A creatine phosphokinase”
Reply: We have changed it.
- line 191 “1_years”
Reply: We have changed it.
- line 195 size of “and”
Reply: We have changed it.
- line 207 please provide Supplementary Table 1
Reply: We have changed “Supplementary Table 1” to “Table 4” and provided it in “Text file”.
- Other change
- In Introduction, line 46-51:
“DEEs are a group of rare, heterogeneous neurodevelopmental disorders characterized by (a) early-onset seizures that are often intractable, (b) electroencephalographic abnormalities, (c) developmental delay or regression, and (d) early death in some patients.2,7,8 EIEE1–EIEE35 are involved in epileptic encephalopathy. In infants with KCNQ2 or STXBP1 encephalopathy, seizures may be controlled early after onset or may cease spontaneously after a few years; however, the developmental consequences are profound.3”
Changed to:
“DEEs are characterized by (a) early-onset seizures that are often intractable, (b) electroencephalographic abnormalities, (c) developmental delay or regression, and (d) early death in some patients [2,7,8]. In infants with KCNQ2 or STXBP1 encephalopathy, seizures may be controlled early after onset or may cease spontaneously after a few years; however, the developmental consequences are profound [3].”
- In Method, Line 82-97
“2.3. WES
DNA for WES was extracted from the peripheral blood of index patients and their parents and was stored at −80°C. For the generation of standard exome capture libraries, we used the Agilent SureSelectXT Reagent Kit protocol for the Illumina Hiseq paired-end sequencing library (catalog#G9611A), and raw reads were mapped to the reference human genome (UCSC Genome Browser-hg19; University of California, Santa Cruz, CA, USA) using BWA version 0.6.5a. The SureSelectXT Human All Exon Version 6 (51 Mb) probe set was used. We used 50 ng genomic DNA to construct the library using the Agilent SureSelectXT Reagent Kit. The amplification adapter-ligated sample was purified using Agencourt AMPure XP beads (Beckman Coulter, Brea, CA, USA) and analyzed using a Bioanalyzer DNA1000 chip. From the purified sample, 750 ng DNA was prepared for hybridization to the capture baits, and the sample was hybridized for 90 min at 65°C using Dynabeads MyOne Streptavidin T1 (Life Technologies, USA) and purified using Agencourt AMPure XP beads. The Agilent protocol was used for adding index tags through posthybridization amplification. Finally, the sample was sequenced on Illumina NextSeq500 by using the 150PE protocol. All samples passed quality control criteria, and more than 99% of the targeted capture region was covered
Changed to:
“2.3. WES
We used the SureSelect XT HS2 DNA reagent kit protocol for the Illumina Hiseq paired-end sequencing library (catalog# G9611A), and raw reads were mapped to the human genome assembly GRCh37 (also known as hg19) using Burrows-Wheeler Aligner version 0.6. The SureSelectXT Human All Exon Version 6 (51 Mb) probe set was used. 50 ng genomic DNA were used for Library Preparation. The adapter-ligated DNA sample was purified using Agencourt Ampure Xp Pcr Purification Beads and analyzed using an Agilent DNA Kit. From the purified sample, the hybridization between DNA libraries (750 ng) and baits were carried out, and purified using Agencourt Ampure Xp Pcr Purification Beads. The Agilent protocol was used for adding tags by Post-Hybridization Amplification. Finally, the sample was sequenced on Illumina NextSeq500 by using the generated reads of 2x150 bp. Every analyte passed all the quality control requirements, and 99% of targeted nucleotides were covered at more than 20X.”
- In conclusion part, line 292
“The findings on the genetic causes of severe childhood neurological disorders are significant. A scientific and efficient WES analysis is particularly important and should be emphasized because the affected gene may not be present in the panel. It highlights the diagnostic relevance of WES for patients with epilepsy in clinical practice.”
Changed to
“The findings on the genetic causes of severe childhood neurological disorders are significant. A scientific and efficient WES analysis is particularly important and should be emphasized because the affected gene may not be present in the panel. It highlights the diagnostic relevance of WES for childhood patients with epilepsy in clinical practice.”
- In Table 1, row “Number of anti-seizure drugs at the time of study” change to “Number of antiepileptic drugs at the time of study”
- We have formatted the reference style to accommodate to Journal of Personalized Medicine.

Reviewer 2 Report
The topic of this manuscript is interesting and valuable. The overall case number was relatively small, so it is a pioneer study which should be indicated in the title. The manuscript provides sufficient background and proper description of the methods. However, the tables and the supplementary were not provided (I cannot find them from the reviewer page) and it is difficult to peer review it without these results. The quality of the figures was not good enough to be published, but these can be modified. I suggest considering the manuscript after a major revision.
Here are some minor points for the modification:
Line 42 please provide exact incidence of the disease.
Line 79 Please provide the details of the Primers for PCR amplification.
Line 98 subtitle number was missing
line 125 All the tables are missing
line 169 A, the letters are too small and not clear. It is recommended to show align figures for all of the sequencing comparison. CD, please make sure the pictures are not deformed. Please also show the scale bar.
line 174 the picture failed to show the feature very well. It is recommended to show local figures to obviously demonstrate the abnormity. Figures from normal children are also recommended to be shown beside the patients’ figures for the comparison.
line 177 A, please enlarge the toenails to be clearer. Figures from normal children are also recommended to be shown beside the patients’ figures for the comparison. C, please point out the corresponding base pair in parental sequences.
line 184 A, the letters are too small and not clear. some figures are not complete (The signal of negative bp were cut out). C, please make sure the pictures are not deformed. Please also show the scale bar.
line 188 size of the “A creatine phosphokinase”
line 191 “1_years”
line 195 size of “and”
line 207 please provide Supplementary Table 1
Author Response
December 6, 2020
Dear Editor-in-Chief and Reviewers:
Manuscript ID: jpm-1029241, Type of manuscript: Article
We are grateful for the opportunity to improve our manuscript and we thank the editorial board and the reviewers for their thoughtful and helpful comments and criticisms. We have modified the paper as suggested. Following are our point-by-point responses. We have also highlighted the principal changes in the “manuscript with the changes marked”
We hope that you, the editorial board, and the reviewers now find our manuscript appropriate for publication in Journal of Personalized Medicine.
We look forward to your reply.
Sincerely yours,
Inn-Chi Lee, MD, PhD
Corresponding Author
The authors analyzed a relatively large number of children with epilepsy, looking for the genetic disorders. They used WES (while-exome sequencing) from blood samples. Genetic diagnoses were confirmed in 24 (out of 57) patients. The study is sound and important, because the genetic diagnosis helps the choice of the appropriate pharmacotherapy. I have a very few suggestions:
- 1. displays a corpus callosum dysgenesis - it seems that the selenium is missing. It should be noted in the Fig. caption.
Reply: Thanks for the valuable opinions. We have changed the figure legend according to reviewers’ suggestions.
In Figure 1, legend:
“Figure 1. Patient 21 has compound heterozygous mutations of SZT2, (A) c.C9055T (p.Arg3019Ter) (arrows) and (B) c.1496+2T>C (arrows), from the asymptomatic mother and father respectively. Seizure onset occurred at 6 months old. His electroencehalogram shows developmental epileptic encephalopathy with multifocal epileptiform discharges. (C) The MRI shows a short and thick corpus callosum (arrow). (D) Her examination was notable for macrocephaly, dysmorphic features, frontal bossing, hypertelorism, microophthalmia, depressed nasal bridge, long-tapered fingers, and hyper-extensible joints.”
Change to:
“Figure 1. Patient 21 has compound heterozygous mutations of SZT2, (A) c.C9055T (p.Arg3019Ter) (arrows) and (B) c.1496+2T>C (arrows), from the asymptomatic mother and father respectively. Seizure onset occurred at 6 months old. His electroencehalogram shows developmental epileptic encephalopathy with multifocal epileptiform discharges. (C) The MRI shows a short and thick corpus callosum with missing of selenium (arrow). (D) Normal child with the same age demonstrates for comparison with patient. (E) Her examination was notable for macrocephaly, dysmorphic features, frontal bossing, hypertelorism, microophthalmia, depressed nasal bridge, long-tapered fingers, and hyper-extensible joints. (F) long-tapered fingers in the patient (red arrow).”
- 2. the ultrasound pictures are of bad quality.
Reply: We have redone Figure 2 according the reviewers’ suggestions. The de novo mutation mean no mutation found in both parents.
“Figure 2. Patient 3 had facial dysmorphic feature, hypotonia, and orogastric tube dependency after birth. (A) His toenails were hypoplastic. Seizure onset occurred at 3 months old. (B) The image of head ultrasound was unremarkable. (C) The WES and sequencing results indicated a novel de novo mutation in the SMARCA4 gene. The arrow indicates a G-to-A substitution at nucleotide 3595 (c.3595G>A, p.Val1199Met) in the patient (arrow), father, and mother. At 2 years, the patient was not able to sit without support.”
Change to:
“Figure 2. Patient 3 had facial dysmorphic feature, hypotonia, and orogastric tube dependency after birth. (A) His toenails were hypoplastic (arrow). Seizure onset occurred at 3 months old. (B) Normal toenails in a healthy infant. (C) The image of head ultrasound was unremarkable. (D) The WES and sequencing results indicated a novel de novo (no mutation in both parents) mutation in the SMARCA4 gene. The arrow indicates a G-to-A substitution at nucleotide 3595 (c.3595G>A, p.Val1199Met) in the patient (arrow), father, and mother. At 2 years, the patient was not able to sit without support.”
- 3. the MRI picture displays sose abnormalities which should be precisely stated in the Fig. caption (microgyria?, myelination problem?, tumor?).
Reply: Thanks for the valuable opinion. The MRI indicates dysmyelination. We have redone the Figure 3 and changed the legend accordingly.
Figure 3. Patient 24 has compound heterozygous mutations of LAMA2, (A) c.1583dupA (p.Ser529GlufsTer19) and (B) c.6931A>T (p.Lys2311Ter), from the asymptomatic mother and father respectively (arrows). (C) His MRI showed basal ganglion, thalamus, white matter and cerebellar vermis signal abnormalities. A creatine phosphokinase was 548 IU/L (reference values, 40-397 IU/L). The genetic study for spinal muscle atrophy was unremarkable. He had developmental delay, recurrent sepsis, and pneumonia with lung atelectasis, and needed respiratory supportive care at home since birth. He died at 1years and 2 months old due to respiratory failure.
Changed to:
“Figure 3. Patient 24 has compound heterozygous mutations of LAMA2, (A) c.1583dupA (p.Ser529GlufsTer19) and (B) c.6931A>T (p.Lys2311Ter), from the asymptomatic mother and father respectively (arrows). (C) His MRI showed basal ganglion, thalamus, white matter and cerebellar vermis signal abnormalities. The finding indicated dysmyelinating process. A creatine phosphokinase was 548 IU/L (reference values, 40-397 IU/L). The genetic study for spinal muscle atrophy was unremarkable. He had developmental delay, recurrent sepsis, and pneumonia with lung atelectasis, and needed respiratory supportive care at home since birth. He died at 1 year and 2 months old due to respiratory failure.”
Comments and Suggestions for Authors in Reviewer 2:
- The topic of this manuscript is interesting and valuable. The overall case number was relatively small, so it is a pioneer study which should be indicated in the title. The manuscript provides sufficient background and proper description of the methods. However, the tables and the supplementary were not provided (I cannot find them from the reviewer page) and it is difficult to peer review it without these results. The quality of the figures was not good enough to be published, but these can be modified. I suggest considering the manuscript after a major revision.
Reply:
- Thanks for the valuable opinion. We have provided the Table 1-3 and Table 4 (old Supplementary Table 1) in the “Text file”
- We changed the “Supplementary Table 1” to “Table 4”.
- We have redone the Figure 1, 2 and 3 according to reviewers’ suggestions.
- In the Title:
We change the title:
“A Wide Spectrum of Genetic Disorders Causing Severe Childhood Epilepsy in Taiwan: A Case-Series of Ultrarare genetic cause and Novel Mutation Analysis” to
“A Wide Spectrum of Genetic Disorders Causing Severe Childhood Epilepsy in Taiwan: A Case-Series of Ultrarare genetic cause and Novel Mutation Analysis in a Pilot Study”
- In line 292, we deleted “Supplementary Materials: The following are available online. Table S1: Diagnostic yield from different methods to identify genetic causes”
Here are some minor points for the modification:
- Line 42 please provide exact incidence of the disease.
In line 42:
“Although the incidence of each disease is low, the combined incidence is high.”
Change to:
“Although the incidence of each disease is low, the combined incidence is not adequately
estimated and unknown.”
- Line 79 Please provide the details of the Primers for PCR amplification.
Reply: Thanks for the opinion. This method is to describe the extracting DNA from Peripheral Blood. Because of this, no primers for PCR were necessary.
In Method:
2.2. Extracting and Amplifying DNA from Peripheral Blood Using PCR
A genomic DNA purification kit (Gentra Puregene Buccal Cell Kit; Qiagen Taiwan, Taipei City) was used to extract genomic DNA from peripheral blood. Briefly, 100 ng genomic DNA was mixed with 10 mM Tris-HCl (pH 9.0), 1.5 mM MgCl2, 50 mM KCl, 0.1% (w/v) gelatin, 1% Triton X-100, 0.2 mM dNTPs, 0.5 mM primers, and 1 unit Taq DNA polymerase (ProTech Technology Enterprise Co., Taipei, Taiwan). The PCR reaction included 35 cycles of 30 s at 94°C, annealing at a special temperature for 30 s, and extension at 72°C for 1 min.
Change to :
“2.2. Extracting DNA from Peripheral Blood
A genomic DNA purification kit (Gentra Puregene Buccal Cell Kit; Qiagen Taiwan, Taipei City) was used to extract genomic DNA from peripheral blood. DNA for WES was extracted from index patients and their parents and was stored at −80°C.”
- Line 98 subtitle number was missing
Reply: We added the subtitle “2.4.”
- line 125 All the tables are missing
Reply: The “Supplementary Table 1” has been changed to “Table 4”
We have provided the Table 1-4 in the Text file.
- line 169 A, the letters are too small and not clear. It is recommended to show align figures for all of the sequencing comparison. CD, please make sure the pictures are not deformed. Please also show the scale bar.
Reply: Thanks for the opinion. We have redone the Figure 1 according to the suggestions.
- line 174 the picture failed to show the feature very well. It is recommended to show local figures to obviously demonstrate the abnormity. Figures from normal children are also recommended to be shown beside the patients’ figures for the comparison.
Reply: We have redone the Figure 1 according to these suggestions.
In Figure 1, legend:
“Figure 1. Patient 21 has compound heterozygous mutations of SZT2, (A) c.C9055T (p.Arg3019Ter) (arrows) and (B) c.1496+2T>C (arrows), from the asymptomatic mother and father respectively. Seizure onset occurred at 6 months old. His electroencehalogram shows developmental epileptic encephalopathy with multifocal epileptiform discharges. (C) The MRI shows a short and thick corpus callosum (arrow). (D) Her examination was notable for macrocephaly, dysmorphic features, frontal bossing, hypertelorism, microophthalmia, depressed nasal bridge, long-tapered fingers, and hyper-extensible joints.”
Change to:
“Figure 1. Patient 21 has compound heterozygous mutations of SZT2, (A) c.C9055T (p.Arg3019Ter) (arrows) and (B) c.1496+2T>C (arrows), from the asymptomatic mother and father respectively. Seizure onset occurred at 6 months old. His electroencehalogram shows developmental epileptic encephalopathy with multifocal epileptiform discharges. (C) The MRI shows a short and thick corpus callosum with missing of selenium (arrow). (D) Normal child with the same age demonstrates for comparison with patient. (E) Her examination was notable for macrocephaly, dysmorphic features, frontal bossing, hypertelorism, microophthalmia, depressed nasal bridge, long-tapered fingers, and hyper-extensible joints. (F) long-tapered fingers in the patient (red arrow).”
- line 177 A, please enlarge the toenails to be clearer. Figures from normal children are also recommended to be shown beside the patients’ figures for the comparison. C, please point out the corresponding base pair in parental sequences.
Reply: Thanks for the valuable opinions. We have redone Figure 2 to be more clear accordingly. The de novo mutation mean no mutation found in both parents.
In Figure 2, legend:
“Figure 2. Patient 3 had facial dysmorphic feature, hypotonia, and orogastric tube dependency after birth. (A) His toenails were hypoplastic. Seizure onset occurred at 3 months old. (B) The image of head ultrasound was unremarkable. (C) The WES and sequencing results indicated a novel de novo mutation in the SMARCA4 gene. The arrow indicates a G-to-A substitution at nucleotide 3595 (c.3595G>A, p.Val1199Met) in the patient (arrow), father, and mother. At 2 years, the patient was not able to sit without support.”
Change to:
“Figure 2. Patient 3 had facial dysmorphic feature, hypotonia, and orogastric tube dependency after birth. (A) His toenails were hypoplastic (arrow). Seizure onset occurred at 3 months old. (B) Normal toenails in a healthy infant. (C) The image of head ultrasound was unremarkable. (D) The WES and sequencing results indicated a novel de novo (no mutation in both parents) mutation in the SMARCA4 gene. The arrow indicates a G-to-A substitution at nucleotide 3595 (c.3595G>A, p.Val1199Met) in the patient (arrow), father, and mother. At 2 years, the patient was not able to sit without support.”
- line 184 A, the letters are too small and not clear. some figures are not complete (The signal of negative bp were cut out). C, please make sure the pictures are not deformed. Please also show the scale bar.
Reply: We have redone the Figure 3 and change the legend accordingly.
“Figure 3. Patient 24 has compound heterozygous mutations of LAMA2, (A) c.1583dupA (p.Ser529GlufsTer19) and (B) c.6931A>T (p.Lys2311Ter), from the asymptomatic mother and father respectively (arrows). (C) His MRI showed basal ganglion, thalamus, white matter and cerebellar vermis signal abnormalities. A creatine phosphokinase was 548 IU/L (reference values, 40-397 IU/L). The genetic study for spinal muscle atrophy was unremarkable. He had developmental delay, recurrent sepsis, and pneumonia with lung atelectasis, and needed respiratory supportive care at home since birth. He died at 1years and 2 months old due to respiratory failure.”
Change to:
“Figure 3. Patient 24 has compound heterozygous mutations of LAMA2, (A) c.1583dupA (p.Ser529GlufsTer19) and (B) c.6931A>T (p.Lys2311Ter), from the asymptomatic mother and father respectively (arrows). (C) His MRI showed basal ganglion, thalamus, white matter and cerebellar vermis signal abnormalities. The finding indicated dysmyelinating process. A creatine phosphokinase was 548 IU/L (reference values, 40-397 IU/L). The genetic study for spinal muscle atrophy was unremarkable. He had developmental delay, recurrent sepsis, and pneumonia with lung atelectasis, and needed respiratory supportive care at home since birth. He died at 1 year and 2 months old due to respiratory failure.”
- line 188 size of the “A creatine phosphokinase”
Reply: We have changed it.
- line 191 “1_years”
Reply: We have changed it.
- line 195 size of “and”
Reply: We have changed it.
- line 207 please provide Supplementary Table 1
Reply: We have changed “Supplementary Table 1” to “Table 4” and provided it in “Text file”.
- Other change
- In Introduction, line 46-51:
“DEEs are a group of rare, heterogeneous neurodevelopmental disorders characterized by (a) early-onset seizures that are often intractable, (b) electroencephalographic abnormalities, (c) developmental delay or regression, and (d) early death in some patients.2,7,8 EIEE1–EIEE35 are involved in epileptic encephalopathy. In infants with KCNQ2 or STXBP1 encephalopathy, seizures may be controlled early after onset or may cease spontaneously after a few years; however, the developmental consequences are profound.3”
Changed to:
“DEEs are characterized by (a) early-onset seizures that are often intractable, (b) electroencephalographic abnormalities, (c) developmental delay or regression, and (d) early death in some patients [2,7,8]. In infants with KCNQ2 or STXBP1 encephalopathy, seizures may be controlled early after onset or may cease spontaneously after a few years; however, the developmental consequences are profound [3].”
- In Method, Line 82-97
“2.3. WES
DNA for WES was extracted from the peripheral blood of index patients and their parents and was stored at −80°C. For the generation of standard exome capture libraries, we used the Agilent SureSelectXT Reagent Kit protocol for the Illumina Hiseq paired-end sequencing library (catalog#G9611A), and raw reads were mapped to the reference human genome (UCSC Genome Browser-hg19; University of California, Santa Cruz, CA, USA) using BWA version 0.6.5a. The SureSelectXT Human All Exon Version 6 (51 Mb) probe set was used. We used 50 ng genomic DNA to construct the library using the Agilent SureSelectXT Reagent Kit. The amplification adapter-ligated sample was purified using Agencourt AMPure XP beads (Beckman Coulter, Brea, CA, USA) and analyzed using a Bioanalyzer DNA1000 chip. From the purified sample, 750 ng DNA was prepared for hybridization to the capture baits, and the sample was hybridized for 90 min at 65°C using Dynabeads MyOne Streptavidin T1 (Life Technologies, USA) and purified using Agencourt AMPure XP beads. The Agilent protocol was used for adding index tags through posthybridization amplification. Finally, the sample was sequenced on Illumina NextSeq500 by using the 150PE protocol. All samples passed quality control criteria, and more than 99% of the targeted capture region was covered”
Changed to:
“2.3. WES
We used the SureSelect XT HS2 DNA reagent kit protocol for the Illumina Hiseq paired-end sequencing library (catalog# G9611A), and raw reads were mapped to the human genome assembly GRCh37 (also known as hg19) using Burrows-Wheeler Aligner version 0.6. The SureSelectXT Human All Exon Version 6 (51 Mb) probe set was used. 50 ng genomic DNA were used for Library Preparation. The adapter-ligated DNA sample was purified using Agencourt Ampure Xp Pcr Purification Beads and analyzed using an Agilent DNA Kit. From the purified sample, the hybridization between DNA libraries (750 ng) and baits were carried out, and purified using Agencourt Ampure Xp Pcr Purification Beads. The Agilent protocol was used for adding tags by Post-Hybridization Amplification. Finally, the sample was sequenced on Illumina NextSeq500 by using the generated reads of 2x150 bp. Every analyte passed all the quality control requirements, and 99% of targeted nucleotides were covered at more than 20X.”
- In conclusion part, line 292
“The findings on the genetic causes of severe childhood neurological disorders are significant. A scientific and efficient WES analysis is particularly important and should be emphasized because the affected gene may not be present in the panel. It highlights the diagnostic relevance of WES for patients with epilepsy in clinical practice.”
Changed to
“The findings on the genetic causes of severe childhood neurological disorders are significant. A scientific and efficient WES analysis is particularly important and should be emphasized because the affected gene may not be present in the panel. It highlights the diagnostic relevance of WES for childhood patients with epilepsy in clinical practice.”
- In Table 1, row “Number of anti-seizure drugs at the time of study” change to “Number of antiepileptic drugs at the time of study”
- We have formatted the reference style to accommodate to Journal of Personalized Medicine.

Round 2
Reviewer 2 Report
The manuscript has been improved significantly. The author address most of my previous concerns except for some problems in the figures. I suggest an acceptance after some minor reversions in the figures.
Here are detailed suggestions:
Line 58 The complete form of abbreviation WES should be presented in which WES first appeared in the main text (besides the abstract).
line 74 Methods of Extracting DNA from Peripheral Blood should be described in detail or refer to a previous study or the manual of the kit.
Line 121 please fill the title row in table 1.
line 122 please note why you bolded some numbers.
line 183 A, the bp letters (CGAT) are too small and not clear. The “ABCD” should be smaller. Please delete the black line on the right side of E.
line 195 AB, please make sure the pictures are not deformed. The scale bar was not clear in B. Please delete the line on the left side of B.
line 203A, the bp letters (CGAT) are too small and not clear. AB, some figures are not complete, the bottom of the figures was cut out. Please replace with the complete figures.
p12 table 4 please center the number of the study in the title row.
Author Response
December 8, 2020
Dear Editor-in-Chief and Reviewers:
Manuscript ID: jpm-1029241, Type of manuscript: Article
We are grateful for the opportunity to improve our manuscript and we thank the editorial board and the reviewers for their thoughtful and helpful comments and criticisms. We have modified the paper as suggested. Following are our point-by-point responses. We have also highlighted the principal changes in the “manuscript with the changes marked”
We hope that you, the editorial board, and the reviewers now find our manuscript appropriate for publication in Journal of Personalized Medicine.
We look forward to your reply.
Sincerely yours,
Inn-Chi Lee, MD, PhD
Corresponding Author
The manuscript has been improved significantly. The author address most of my previous concerns except for some problems in the figures. I suggest an acceptance after some minor reversions in the figures.
Here are detailed suggestions:
- Line 58 The complete form of abbreviation WES should be presented in which WES first appeared in the main text (besides the abstract).
Reply: In line 59:
“The aim of this study was to determine the diagnostic utility of WES for a heterogeneous group of childhood DEEs and to develop a viable protocol for identifying severe neurological disease.”
Changed to
“The aim of this study was to determine the diagnostic utility of whole-exome sequencing (WES) for a heterogeneous group of childhood DEEs and to develop a viable protocol for identifying severe neurological disease.”
- line 74 Methods of Extracting DNA from Peripheral Blood should be described in detail or refer to a previous study or the manual of the kit.
Reply: In line 75:
2.2. Extracting DNA from Peripheral Blood
A genomic DNA purification kit (Gentra Puregene Buccal Cell Kit; Qiagen Taiwan, Taipei City) was used to extract genomic DNA from peripheral blood. DNA for WES was extracted from index patients and their parents and was stored at −80°C.
Changed to:
“A genomic DNA purification kit (Gentra Puregene Buccal Cell Kit; Qiagen Taiwan, Taipei City) was used to extract genomic DNA from peripheral blood. Three volumes of RBC lysis buffers were added to blood sample and mixed by inverting 30 times and incubated for 10 minutes at room temperature and centrifuged 3000 rpm for 5 minutes. The supernatant was discarded. Add 3 volumes of cell lysis buffers (3ml) to the pellet and vortex vigorously for 10 seconds to lyse the cell and centrifuged 3000 rpm for 5 minutes, followed by transferring the supernatant into a new 15 ml tube and added 3 ml isopropanol into a 15ml tube and mixed by inverting 50 times, and the centrifuged 3000 rpm for 5 minutes. The supernatant was carefully discard and air dry the pellet for 10 to 20 minutes and the dried pellet was resuspended in 300 μl nuclease-free water and frozen at −20°C or −80°C for storage. DNA were also extracted from their parents.”
- Line 121 please fill the title row in table 1.
Reply: In Row 1 of Table, we added “ Clinical data (N)”
- line 122 please note why you bolded some numbers.
Reply: We have corrected it in Figure legends.
In Figure 1, legend: “Figure 1. Figure 1.” changed to “ Figure 1.”
- line 183 A, the bp letters (CGAT) are too small and not clear. The “ABCD” should be smaller. Please delete the black line on the right side of E.
Reply: We have redone Figure 1 and corrected these problems.
In Figure 1, legend: line 165:
“….from the asymptomatic mother and father respectively.…” chaned to “….from the asymptomatic father and mother respectively.”
- line 195 AB, please make sure the pictures are not deformed. The scale bar was not clear in B. Please delete the line on the left side of B.
Reply: We have redone Figure 2 and corrected these problems.
- line 203A, the bp letters (CGAT) are too small and not clear. AB, some figures are not complete, the bottom of the figures was cut out. Please replace with the complete figures.
Reply: We have redone Figure 3 accordingly.
- p12 table 4 please center the number of the study in the title row.
Reply: We have changed it in Table 4.
